# The Influence of Mindful Parenting on Children’s Creative Tendencies: The Chain Mediating Effect of Parent–Child Intimacy and Connectedness to Nature

**DOI:** 10.3390/bs14030223

**Published:** 2024-03-10

**Authors:** Jingyu He, Ziming Wang, Yue Zhang, Shuo Feng, Jinxia Han, Hehong Quan, Chun Li

**Affiliations:** 1Department of Preschool Education, Qingdao University, Qingdao 266071, China; hejingyu@qdu.edu.cn (J.H.); wangziming@qdu.edu.cn (Z.W.); zhangyue65@qdu.edu.cn (Y.Z.); fengshuo@qdu.edu.cn (S.F.); 2Department of Education, Qufu Normal University, Qufu 273165, China; hanjinxia@qfnu.edu.cn; 3Graduate School of Humanities and Social Sciences, Hiroshima University, Hiroshima 7398511, Japan

**Keywords:** mindful parenting, children’s creative tendencies, parent–child intimacy, connectedness to nature

## Abstract

(1) Objectives: The aim of this study was to examine the relationship between mindful parenting and children’s creative tendencies and to investigate the mediating role of parent–child intimacy and connectedness to nature in the relationship between mindful parenting and children’s creative tendencies. (2) Methods: In this cross-sectional study, nearly 800 mothers of children aged 3–6 were enrolled. General sociodemographic data, the Mindfulness in Parenting Questionnaire (MIPQ), the Creativity Assessment Packet (CAP), the Child–Parent Relationship Scale—Short Form (CPRS-SF), and the Connectedness to Nature Index—Parents of Preschool Children (CNI-PPC) were all included in the questionnaire survey. (3) Results: There were significant positive correlations among mindful parenting, parent–child intimacy, connectedness to nature, and children’s creative tendencies. Mindful parenting had a positive predictive effect on children’s creative tendencies. Parent–child intimacy played a mediating role between mindful parenting and children’s creative tendencies. Connectedness to nature played a mediating role between mindful parenting and children’s creative tendencies. The correlation between mindful parenting and children’s creative tendencies may be impacted by the chain mediation effects of parent–child intimacy and connectedness to nature. (4) Conclusions: By promoting parent–child intimacy and connectedness to nature, and by utilizing the chain mediating effects of both, mindful parenting positively impacted children’s creative tendencies.

## 1. Introduction

Childhood is a critical period for developing creativity, and it has a large impact on an individual’s creative evolution throughout their lives [1]. According to research, the family environment is one of the most important sources of creative development in preschool children [2]. According to the ecosystem model of creative development, parenting approaches have an impact on creativity [3]. Mothers’ favorable parenting styles predict their children’s creative talents [4]. Several studies have recently confirmed mindful parenting as a positive and effective parenting style that can benefit children, parents, and parent–child relationships, such as assisting children in reducing internalizing and externalizing problems [5], promoting children’s social competence [6], alleviating parenting stress [7], improving parenting skills [8], and improving parent–child relationships [9], among other things. However, there is no direct evidence that mindful parenting promotes creativity in preschool children, particularly the tendency to be creative, which is an important psychological barrier to creativity development and performance [10]. Based on this, it is crucial to explore in depth the mechanisms by which mindful parenting influences preschoolers’ creative tendencies.

Mindful parenting is the ability of parents to deliberately incorporate a present-centered, nonjudgmental awareness into their daily interactions with their children [11]. In the words of Duncan et al. (2009), mindful parenting includes five essential elements in the interpersonal process of raising children: active listening, nonjudgmental acceptance of oneself and one’s child, emotional awareness of oneself and the child, self-regulation in the parent–child relationship, and compassion for oneself and the child. The development of children’s autonomy as well as the psychological health and positive parenting styles of parents can both benefit from mindful parenting. It might also lessen parents’ rejection and excessive control over their kids [12]. When Kathleen et al. (1973) looked into the parents of kids who had many creative tendencies, they discovered that these parents had positive parenting styles that included valuing equality, emphasizing self-awareness, letting their kids express themselves on their own, and patiently listening to their kids’ ideas [13,14]. Parents who practice mindful parenting also respect their children’s needs and ideas in their daily lives, and they respond to their children’s behaviors with warmth and support, which is consistent with the demonstration of a positive parenting style, even though there is no direct evidence to support the idea that mindful parenting influences preschoolers’ creative tendencies [15]. To find out whether mindful parenting may influence children’s creative instincts or whether it can only encourage them, more research is necessary. Simultaneously, a number of research works have demonstrated that mothers exhibit higher levels of mindful parenting than fathers, given that they are the primary caregivers of preschool-aged children [4,16,17]. And, in daily life, the mother is the main person who creates a creative environment [18]. As a result, the current research indicates that mindful parenting practices and mothers’ encouraging reactions to their kids may be crucial in influencing kids’ creative inclinations. Based on the research above, this study proposes Hypothesis 1—mindful parenting can positively predict children’s creative tendencies.

Ecological systems theory suggests that the family is a significant interpersonal environment that influences individual development, and parent–child intimacy is one important variable within it [19]. Research has shown that mindful parenting can positively predict parent–child intimacy [20,21], assist parents in accepting and eliminating conflict in the parent–child relationship [22], and so foster warm emotional support and healthy parent–child communication. Some findings also suggest that maintaining a close parent–child bond within a supportive family environment can significantly increase children’s creative potential and contribute to the shaping of their creative tendencies [23]. Parents who practice mindful parenting are very good at creating a supportive and warm family atmosphere [24]. They provide more security for their children, empower them to make their own decisions, encourage their children to experience novel activities and experiment with diverse behaviors, and are more aware of children’s creative expressions compared to other parents [25,26]. Therefore, this study proposes Hypothesis 2—parent–child intimacy mediates the relationship between mindful parenting and children’s creative tendencies.

Nature is the best teacher for children rather than books and lectures [27]. Many experts feel that regular exposure to nature in the early and middle preschool years fosters positive views about the natural world, resulting in strong environmental linkages [28,29]. Connectedness to nature refers to an intimate interaction between humans and nature in which people are emotionally attached to nature, cognitively blend nature with themselves, and physically experience nature’s allure and are willing to coexist with it [30]. According to Nicholson’s (1973) idea of loose parts, including loose parts into natural environments provides children with numerous possibilities for play, which supports the development of children’s creative tendencies [31,32]. Furthermore, creatures, plants, and ecosystems in natural settings can entice children to actively investigate and encourage them to switch from one interest to another, encouraging them to explore and create [33]. These exploratory experiences can also help children stimulate their imaginations when in close contact with nature [34] and foster interest about the outer world [35]. In addition, parents who practice positive parenting use democratic and enlightened positive parenting [20] and support children’s exploration of the natural environment outside, thus helping them develop good natural connections [36]. Based on this, Hypothesis 3 is proposed—connectedness to nature mediates the relationship between mindful parenting and children’s creative tendencies.

The natural environment has a distinct favorable effect on parent–child relationships [37,38]. Research has shown that children can fully engage in communication with nature in the natural settings provided by nature with the help of a close parent–child relationship, increasing their level of connection to nature through a close and balanced dialogue with nature [39]. In order to establish a more equal and close relationship with their children, parents can interact with them in the context of mindful parenting by remaining emotionally neutral and accepting of them. In turn, this closeness between parents and children can encourage children’s natural curiosity and imagination, as well as assist them in progressively developing a deeper connection to nature [40]. Based on the above, Hypothesis 4 is proposed—parent–child intimacy and connectedness to nature play a chain mediating role in the effect of mindful parenting on children’s creative tendencies in a chain mediation model.

In conclusion, mothers who practice mindful parenting can demonstrate greater flexibility and responsiveness, which tends to improve parent–child intimacy and help young children increase their level of connectedness to nature, influencing the shaping of preschoolers’ creative tendencies. Considering these factors, this study constructed a chain mediation model based on ecosystem theory, aiming to validate the multiple mediating roles of parent–child intimacy and the connection to nature between mindful parenting and children’s creative tendencies. The concept of chain mediation refers to the sequential properties shared by several mediating variables that constitute a chain of mediators. Through this chain, the predictor variable has an indirect effect on the result variable [41]. Such a study might possibly increase our understanding of mindful parenting for preschool children’s development while also providing important lessons and insights for education and practices for the development of children’s creativity.

## 2. Materials and Methods

### 2.1. Participants

Convenience sampling was used in this study, and questionnaires were distributed to parents of preschool children in six kindergartens in Shandong Province, China, including kindergarten classes for ages 3–4 (n = 316), ages 4–5 (n = 231), and ages 5–6 (n = 294). A total of 1126 questionnaires were distributed, and 285 incomplete or irregularly answered questionnaires were excluded from the analysis. The exclusion criteria were as follows: (1) missing data in the returned questionnaires; (2) the same answers for 80% of the items in the returned questionnaires; and (3) time to complete the questionnaires <10 min. The final valid sample consisted of 841 participants: 74.7% including 412 boys (49.0%) and 429 girls (51.0%).

### 2.2. Procedure

The researcher recruited mothers from six kindergartens using a convenience sampling technique, and then gave research questionnaires to them via the kindergartens. The questionnaires, which took between 15 and 20 min to complete, were filled out voluntarily by the participants. All the questionnaires were then gathered by the researcher.

### 2.3. Measures

#### 2.3.1. Demographic Information

The questionnaire included a demographic information sheet, which collected information such as whether the participant’s child was an only child, the participant’s occupation, education level, and monthly family income, as shown in Table 1.

#### 2.3.2. Mindful Parenting

Mindful parenting was assessed with the Chinese version [42] of the Mindfulness in Parenting Questionnaire (MIPQ) [43]. The scale includes two dimensions: present moment attention and mindful discipline, comprising a total of 28 items scored on a 4-point Likert scale ranging from 1 (never) to 4 (always). Mothers were asked to choose a descriptive level among the entries that best fit them as they had actually been in the last month (including the day in question). Present moment attention (13 items) refers to the child-oriented aspects of mindfulness parenting, including present-focused attention, empathy, and acceptance of the child, such as “Did you carefully listen and tune into your child when you two were talking”; and the focus on mindful discipline (15 items) refers to the parent-oriented aspects of mindfulness parenting, including non-reactivity in parenting, parenting awareness, and goal-directed parenting, such as “Did you believe that the way you were parenting was consistent with best parenting practices”. A higher total score indicates a higher frequency of mindful parenting behavior. In this measurement, the overall Cronbach’s α coefficient of the scale was 0.95, and the Cronbach’s α coefficients for the dimensions of present-moment attention and mindfulness discipline were 0.92 and 0.93.

#### 2.3.3. Children’s Creative Tendencies

Children’s creative tendencies were assessed with the Chinese version [44] of the Creativity Assessment Packet (CAP) [45]. The scale consists of 50 items grouped into four subscales: adventure, curiosity, imagination, and challenge. Parents who had knowledge of their preschool children’s situation were asked to complete the questionnaire based on their child’s behavior in the past six months. Responses were scored on a 4-point Likert scale ranging from 1 (completely inconsistent) to 4 (completely consistent). Adventurousness (11 items), such as “My child likes to try to guess things or problems in school, even if it is not necessarily correct”; curiosity (14 items), like “My child likes to observe things he has not seen before to learn about details”; imagination (13 items), like “My child likes to imagine things he wants to know or do”; and challenge (12 items), such as “My child likes to listen to unpredictable and imaginative stories” were evaluated. A higher score indicated stronger creative tendencies in preschool children. In this study, the overall Cronbach’sα coefficient of the scale was 0.93, and the Cronbach’s α coefficients for the dimensions of adventurousness, curiosity, imagination, and challenge were 0.71, 0.78, 0.80, and 0.75.

#### 2.3.4. Parent–Child Intimacy

Parent–child intimacy was assessed with the Chinese version [46] of the Child–Parent Relationship Scale—Short Form (CPRS-SF) [47]. The measure was filled out by mothers to evaluate their interactions with their children. The scale, comprising 15 items grouped into two dimensions of intimacy and conflict, is aimed at measuring the degree of closeness and harmony in parent–child relationships (7 items), like “I share an affectionate, warm relationship with my child”. The conflict dimension (8 items) mainly involves conflicts or deteriorations in the parent–child relationship, such as “My child and I always seem to be struggling with each other”. Responses are scored on a 5-point Likert scale ranging from 1 (completely inconsistent) to 5 (completely consistent). In this study, the intimacy dimension of the scale was used to predict parent–child intimacy, with a higher score indicating a closer parent–child relationship. The Cronbach’s alpha coefficient for the scale was 0.79 in this study.

#### 2.3.5. Connectedness to Nature

Connectedness to nature was evaluated using the Connectedness to Nature Index—Parents of Preschool Children (CNI-PPC) [48]. It was used to assess preschool children’s level of bonding with nature by measuring four dimensions: enjoyment of nature, empathy for nature, responsibility toward nature, and awareness of nature. Enjoyment of nature (6 items) refers to the child’s liking for being in contact with nature and feeling comfortable and happy in nature, such as “My child likes to hear different sounds in nature”; empathy for nature (3 items) refers to the children’s emotional response when animals or plants are hurt or die, like “My child feels sad when wild animals are hurt”; responsibility toward nature (3 items) refers to the child’s behavior of protecting the environment, for example “My child believes that picking up trash on the ground can help the nature”; and awareness of nature (4 items) refers to the children’s perception and understanding of nature in various forms, like “My child notices wildlife wherever he/she is”. The scale consists of 16 items scored on a 5-point Likert scale ranging from 1 (disagree) to 5 (agree). A higher score indicates stronger bonding with nature in preschool children. In this study, the overall Cronbach’s α coefficient of the scale was 0.93, and the Cronbach’s α coefficients for the dimensions of enjoyment of nature, empathy for nature, responsibility toward nature, and awareness of nature were 0.87, 0.87, 0.68, and 0.81.

### 2.4. Data Analysis

Data analysis in this study was conducted using SPSS 24.0. As recommended by Podsakoff et al., we first checked for common method bias in the study [49]. Second, descriptive statistics and correlation analyses were then conducted on the data. Finally, we tested for the chain mediation effect using Hayes’ PROCESS macro (Model 6), while controlling for covariates.

## 3. Results

### 3.1. Common Method Bias Testing

As all study variables in this research were measured using maternal self-report questionnaires, there might have been a potential for common method bias in the data. Thus, it was necessary to check for the presence of common method effects before conducting data analysis to determine whether they would significantly affect the study results. According to Zhou and Long’s (2004) recommendation, we controlled for common method effects by using anonymous questionnaires during data collection [50]. To further validate the rigor of this study, we performed a Harman’s single-factor test to detect common method bias [49]. The results showed that 22 factors with eigenvalues greater than 1 were extracted, and the variance explained by the first factor was 23.26%, which is less than the criterion of 40%, indicating that there was no serious common method bias present in the data of this study.

### 3.2. Preliminary Analysis

Table 2 presents the mean, standard deviation, and correlation matrix of the study variables. The results indicate significant positive correlations among mindfulness parenting, parent–child intimacy, connectedness to nature, and children’s creative tendencies. Specifically, mindful parenting, parent–child intimacy, and children’s creative tendencies were significantly positively correlated with all four dimensions of connectedness to nature, including enjoyment of nature, empathy for nature, awareness of nature, and responsibility toward nature.

### 3.3. Multiple Mediating Model Analysis

Based on the results above, we used demographic variables such as gender, grade, parental occupation, parental education level, and family income as control variables in our mediation model. Parent–child intimacy and connectedness to nature were used as mediator variables. The mediation model was tested using SPSS macro PROCESS (Model 6) with 5000 bootstrap samples, and the results are presented in Table 3 and Figure 1. Mindfulness parenting significantly predicted children’s creative tendencies (*c*), parent–child intimacy (*a*_1_), and connectedness to nature(*a*_2_). Parent–child intimacy significantly predicted connectedness to nature (*d*) and children’s creative tendencies (*b*_1_). Connectedness to nature also significantly predicted children’s creative tendencies (*b*_2_).

Using mindful parenting as the independent variable and children’s creative tendencies as the dependent variable, while controlling for demographic variables such as gender, grade, parental occupation, parental education level, and family income, and using parent–child intimacy and connectedness to nature as mediator variables, a mediation analysis was conducted, as presented in Table 4 and Figure 1. The results indicate significant mediation effects of parent–child intimacy and connectedness to nature on the relationship between mindful parenting and children’s creative tendencies (total indirect effect = 0.155, SE = 0.021, bootstrap 95% CI: [0.114, 0.200]). The proportion of this effect to the total effect (0.434) was 35.71. At the same time, there was a significant direct effect path (*c’*) between mindful parenting and children’s creative tendencies (direct effect = 0.279, SE = 0.032, bootstrap 95% CI: [0.000, 0.216]). This indicates that parent–child intimacy and connectedness to nature partially mediate the relationship between mindful parenting and children’s creative tendencies.

The mediation effect consists of three indirect paths: First, the independent mediation effect of parent–child intimacy: Mindful parenting → Parent–child intimacy → Children’s creative tendencies (*a*_1_**b*_1_, mediating effect = 0.060, SE = 0.020, bootstrap 95% CI: [0.022, 0.101], accounting for 13.82 of the total effect); Second, the independent mediation effect of connectedness to nature: Mindful parenting → Connectedness to nature → Children’s creative tendencies (*a*_2_**b*_2_, mediating effect = 0.052, SE = 0.013, bootstrap 95% CI: [0.029, 0.078], accounting for 11.98 of the total effect); and Third, the chain mediation effect of parent–child intimacy and connectedness to nature: Mindful parenting → Parent–child intimacy → Connectedness to nature → Children’s creative tendencies (*a*_1_**d*b*_2_, mediating effect = 0.043, SE = 0.008, bootstrap 95% CI: [0.030, 0.059], accounting for 9.91 of the total effect). The confidence intervals for each path do not include zero, indicating that the chain mediation is established, and the mediation effect is significant.

## 4. Discussion

### 4.1. The Effect of Mindful Parenting on Children’s Creative Tendencies

This study verified Hypothesis 1 by demonstrating that mothers’ mindful parenting can accurately predict their children’s creative tendencies. Previous studies have less commonly investigated the association between mindful parenting and creativity. The data collected from 841 mothers’ reports in this study confirmed the positive relationship between mindful parenting and preschoolers’ creative tendencies. A recent study of adolescents aged 12–15 years and their parents discovered that the higher the parents’ degree of mindful parenting, the better the development of their children’s creativity and their level of creativity in all its dimensions [51]. Our findings support previous research, broaden the age range of existing studies, and give additional empirical evidence for shaping children’s creative tendencies. We contend that the findings of this study may be linked to mothers’ habits of relating to their children. According to several research works, mothers are more able to provide their children the freedom to express their thoughts, feelings, and wants throughout their time with them, helping them learn to respond and encouraging them to develop independent thinking, responsibility, and self-confidence [52,53]. According to Setiyowati et al. (2019), more than half of the children in a given sample will have a high level of creative development in a democratic home environment provided by the mother for the child [54]. Mothers who practice mindful parenting can listen attentively to their children’s needs, accept their ideas without judgment, respond positively to their children’s behaviors, and so on when they are with them. This can also provide children with a respectful and supportive family environment where they are treated as equals [55,56]. Thus, children who feel more supported and secure with their moms are more likely to explore freely, which fosters the development of their creative tendencies [57]. These findings also support earlier beliefs.

### 4.2. The Chain-Mediated Effect of Parent–Child Intimacy and Connectedness to Nature on the Relationship between Mindful Parenting and Children’s Creative Tendencies

This study discovered that parent–child intimacy was an independent mediating factor in the association between mindful parenting and children’s creative tendencies, confirming Hypothesis 2. This result is consistent with previous studies that mindful parenting helps enhance parent–child intimacy as well as healthy parent–child affection during encounters [58]. When children experience their mothers’ positive emotional support, they gradually begin to let down their psychological barriers to external stimuli, develop a sense of challenge and curiosity, and display bravery in facing challenges and difficulties [59]. Myers (1997) thought that there was a reciprocal relationship between adult emotional reactions and children’s emotional reactions [60]. Mothers choose to engage in meaningful interactions with their children to bridge the gap between them and themselves through proactive and positive emotions, which in turn enhances the child’s positive emotional reaction in a “gear-turning” manner. In some circumstances, this may improve the synchronization of parent–child interactions and strengthen the link between them [61]. Positive emotions enhance young children’s cognitive flexibility, and their language, conduct, play, and other behaviors demonstrate vivid imagination, dramatic perception, imagery, and association skills [62].

The findings of this study also indicated that connectedness to nature plays an independent mediation role in the relationship between mindful parenting and children’s creative tendencies, supporting Hypothesis 3. It has been established that if parents have a more positive attitude toward the natural environment, i.e., they are willing to experience nature with their children or allow them to play in nature [63], their children will develop a unique emotional attachment to nature, a sense of responsibility to nature, and a willingness to explore and take risks in nature [64,65]. This high level of connection to nature in children contributes to the development of long-term, cross-cutting creative tendencies that foster children’s creative achievement [40]. The findings corroborate and expand on earlier research. In other words, mothers who practice mindful parenting allow their children to go outside and engage with nature more frequently. Every flower and tree in nature contributes to the development of creative talents such as imagination and exploration, hence developing the child’s creative tendencies.

Notably, multiple research projects have found a substantial link between high levels of mindfulness in people and exposure to nature [66,67]. High levels of mindfulness may help mothers better embrace nature through the trait of non-judgment by restoring the ability to pay attention during encounters with the natural environment [39]. Mothers are also better able to focus on the present moment, become aware of their own and their children’s feelings and ideas, and improve their mindful parenting skills in the process [56]. Children have a natural yearning to be in intimate contact with nature [68]. Research has indicated that being exposed to nature, such as through outdoor play or recreation, is an excellent approach to fostering a connection to nature [69]. This study also discovered that children’s need to yearn for nature was more likely to be perceived by mothers who practiced mindful parenting and were willing to take the initiative to accompany their children in experiencing nature and allowing them to play in nature with a more open mind [70]. This not only allows children to acquire high levels of organically related behaviors, but it also subconsciously instills endogenous incentives in the development of children’s creative tendencies [71].

Finally, parent–child intimacy and natural connection mediated the association between mindful parenting and children’s creative tendencies, supporting Hypothesis 4. Sen and Sharma (2013) argued that creativity is an interactive and dynamic relationship between people and their environments, rather than a product of a certain setting [72]. The outcomes of this study lend support to the ecological systems theory, which holds that human interaction with their environment leads to development [19]. From this standpoint, both mother-child interaction and a sense of connection to nature may influence how creatively children develop. Mothers can improve parent–child relationships and closeness between parents and children by practicing mindful parenting. Children can transfer their positive emotional experiences by having harmonious interactions with their mothers, enabling children to love nature and enjoy nature. As children develop their own cognitive and emotional identification with nature, their bond with nature strengthens and is thus more likely to facilitate creative problem-solving [73], resulting in practical benefits for children’s creative tendencies.

### 4.3. Limitations and Directions for Future Research

Some limitations of this study should be addressed in future research. First and foremost, the data used in this study were limited to mothers’ reports. Most prior research on mindful parenting has mostly included mothers [74,75]. However, mothers and fathers play distinct roles in the parenting process and parent–child connection [76,77]. In addition, studies have demonstrated that dads’ parenting styles can promote the development of children’s creativity [78]. As a result, future research could use dads as the focus of the report to investigate the effects on children’s creative tendencies.

Secondly, the importance of children’s creativity tendencies in forecasting and guiding their creative behaviors deserves to be examined and investigated in depth, and while children were the major participants in this study, they were not involved in the research. Future studies should incorporate more experimental ways to investigate children’s actual creative performance.

Finally, the Natural Connections Inventory (CNI-PPC) employed in this study was developed for Hong Kong school-aged children, and further research is required to evaluate its reliability and validity when measured in Mainland Chinese children.

### 4.4. Theoretical and Practical Contributions

On a theoretical level, this study developed a chain mediation model based on ecological systems theory, which supports the impact of mindful parenting on children’s creative tendencies as well as the chain-mediated function of parent–child intimacy and connection to nature in the relationship between the two. The study’s findings added to and enlarged the applicable ideas of mindful parenting and children’s creativity, providing a useful platform for future research.

At the practical level, the findings of this study provide new directions for educational activities aimed at developing children’s creative tendencies. Parents should use mindful parenting approaches to encourage their children’s creative tendencies. These methods include accepting their children’s conduct without passing judgment, focusing on their ideas and emotional needs, paying attention to their own emotional experience to build parent–child intimate relationships, and intentionally expanding the time and space for parent–child connection. Parents can also take their kids outside to explore nature. Furthermore, kindergartens can help children develop their creative tendencies by organizing parent–child activities that stimulate natural experiences or actively lobbying for appropriate educational concepts with parents.

## 5. Conclusions

This study found that mindful parenting had a positive predictive effect on children’s creativity tendencies. Both parent–child intimacy and connectedness to nature play an independent mediating role between mindful parenting and children’s creative tendencies. Parent–child intimacy and connectedness to nature play a chain mediating role between mindful parenting and children’s creative tendencies.

## Figures and Tables

**Figure 1 behavsci-14-00223-f001:**
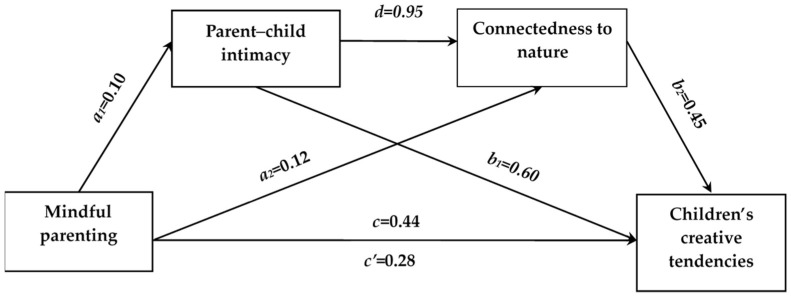
The pathway of the mediating model.

**Table 1 behavsci-14-00223-t001:** Demographic characteristics for the participants (n = 841).

Characteristics (Code)	n (%)
Child’s gender	
Boy	412 (49.0)
Girl	429 (51.0)
Child’s grade	
Children aged 3–4 years old	316 (37.6)
Children aged 4–5 years old	231 (27.5)
Children aged 5–6 years old	294 (35.0)
Parent’s occupation	
A professional	32 (3.8)
A company’s administrator	246 (29.3)
A company’s employee	256 (30.4)
A service provider	57 (6.8)
A worker	27 (3.2)
Self-employed	107 (12.7)
Unemployed	77 (9.2)
Other	39 (4.6)
Only child	
Yes	316 (37.6)
No	525 (62.4)
Education level of the parent	
Junior high school and below	11 (1.3)
Senior high school	69 (8.2)
Junior college	169 (20.1)
Bachelor’	382 (45.4)
Master	154 (18.3)
Doctor	56 (6.7)
Average monthly family income	
Below 5000	50 (5.9)
5000~10,000	184 (21.9)
10,000~15,000	208 (24.7)
15,000~20,000	151 (18)
More than 20,000	248 (29.5)

**Table 2 behavsci-14-00223-t002:** Descriptive statistics and correlation analysis.

	M	SD	1	2	3	4	5	6	7	8
1 Mindful parenting	86.80	13.10	1							
2 Parent–child intimacy	31.38	3.13	0.42 ***	1						
3 Connectedness to nature	72.19	8.78	0.31 ***	0.41 ***	1					
4 Enjoyment of nature	27.63	3.40	0.27 ***	0.41 ***	0.91 ***	1				
5 Empathy for nature	13.55	2.05	0.24 ***	0.32 ***	0.83 ***	0.65 ***	1			
6 Awareness of nature	18.18	2.35	0.30 ***	0.40 ***	0.91 ***	0.77 ***	0.69 ***	1		
7 Responsibility toward nature	12.82	2.16	0.29 ***	0.29 ***	0.86 ***	0.67 ***	0.67 ***	0.73 ***	1	
8 Children’s creative tendencies	113.71	13.13	0.43 ***	0.38 ***	0.45 ***	0.41 ***	0.36 ***	0.42 ***	0.39 ***	1

Note: *** *p* < 0.001.

**Table 3 behavsci-14-00223-t003:** Multiple regression analysis.

Regression Model		Regression Coefficient Significance				Overall Fitting Index		
Dependent variable	Independent variable	**β**	**t**	**LLCI**	**ULCI**	**R**	**R^2^**	**F**
Children’s creative tendencies	Gender	1.97	2.40	0.36	3.57	0.44	0.20	29.20 ***
	Grade	0.96	1.96 *	0.00	1.92			
	Only child	−0.81	−0.94	−2.50	0.88			
	Parent’s occupation	−0.14	−0.31	−1.01	0.73			
	Parent’s educational level	−0.14	−0.63	−0.59	0.30			
	Average monthly family income	0.06	0.16	−0.61	0.71			
	Mindful parenting	0.44	13.67 ***	0.37	0.50			
Parent–child intimacy	Gender	0.33	1.68	−0.06	0.71	0.44	0.19	28.02 ***
	Grade	−0.02	−0.19	−0.25	0.21			
	Only child	−0.62	−3.02 *	−1.03	−0.22			
	Parent’s occupation	0.02	0.20	−0.18	0.23			
	Parent’s educational level	0.04	0.77	−0.07	0.15			
	Average monthly	0.15	1.80	−0.01	0.30			
	Mindful parenting	0.10	12.69 ***	0.08	0.11			
Connectedness to nature	Gender	2.04	3.75 ***	0.97	3.10	0.46	0.21	28.03 ***
	Grade	0.45	1.37	−0.19	1.08			
	Only child	0.53	0.93	−0.59	1.66			
	Parent’s occupation	−0.47	−1.61	−1.05	0.10			
	Parent’s educational level	−0.19	−1.24	−0.48	0.11			
	Average monthly	0.07	0.32	−0.37	0.51			
	Mindful parenting	0.12	5.02 ***	0.07	0.16			
	Parent–child intimacy	0.95	9.90 ***	0.76	1.14			
Children’s creative tendencies	Gender	0.72	0.95	−0.78	2.22	0.56	0.32	42.48 ***
	Grade	0.78	1.73	−0.10	1.67			
	Only child	−0.41	−0.51	−1.98	1.16			
	Parent’s occupation	0.05	0.12	−0.75	0.85			
	Parent’s educational level	−0.10	−0.49	−0.52	0.31			
	Average monthly	−0.13	−0.40	−0.74	0.49			
	Mindful parenting	0.28	8.73 ***	0.22	0.35			
	Parent–child intimacy	0.60	4.24 ***	0.32	0.88			
	Connectedness to nature	0.45	9.21 ***	0.35	0.54			

Note: *** *p* < 0.001. * *p* < 0.05.

**Table 4 behavsci-14-00223-t004:** Indirect effects of parent–child intimacy and connectedness to nature.

	Effect	Boot SE	Boot LLCI	Boot ULCI	Ratio of Total
Total indirect effect	0.155	0.021	0.114	0.200	35.71%
Indirect effect 1 (*a_1_*b_1_*)	0.060	0.020	0.022	0.101	13.82%
Indirect effect 2 (*a_2_*b_2_*)	0.052	0.013	0.029	0.078	11.98%
Indirect effect 3 (*a_1_*d*b_2_*)	0.043	0.008	0.030	0.059	9.91%

## Data Availability

No new data were created for this study.

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
