# Peer review of "The Influence of Mindful Parenting on Children’s Creative Tendencies: The Chain Mediating Effect of Parent–Child Intimacy and Connectedness to Nature"

_behavsci, 2024, doi:10.3390/bs14030223_

Round 1

Reviewer 1 Report

Comments and Suggestions for Authors

The aim of this study was to explore the relationship between mindful parenting and children's creative tendency, and to investigate the mediating role of parent-child intimacy and connectedness to nature in this relationship. The results revealed significant positive correlations among mindful parenting, parent-child intimacy, connectedness to nature, and children's creative tendency. Mindful parenting had a positive predictive effect on children's creative tendency. Parent-child intimacy acted as a mediator between mindful parenting and children's creative tendency. Similarly, connectedness to nature served as a mediator between mindful parenting and children's creative tendency. The correlation between mindful parenting and children's creative tendency might be influenced by the chain mediation effects of parent-child intimacy and connectedness to nature. This study contributes substantively to both theory and practice. However, the following recommendations are suggested for the authors to consider:

1. In the Introduction, it is advisable for the authors to review past literature related to this study, highlighting the limitations, research questions, or gaps identified in previous studies. How does this study address previously unanswered questions? Additionally, how does it contribute to theory or practice?

2. On page 3, a total of 1126 questionnaires were distributed, with 285 excluded due to incompleteness or irregular responses. Please specify the exact criteria or method used for excluding these questionnaires.

3. On page 4, when explaining the MIPQ, the description mentions two dimensions: present moment attention and mindful discipline. However, the Cronbach's alpha coefficient is presented only as an overall coefficient. Please clarify why the alpha coefficient is only presented for the overall scale and not for each dimension.

4. On page 5, regarding the Chinese version of the Creativity Assessment Packet (CAP) with four subscales, only one Cronbach's alpha coefficient is presented. It's recommended to provide the Cronbach's alpha coefficient for each subscale. Similar discrepancies in reporting alpha coefficients for other scales during their explanation also need to be addressed.

5. In Table 2 on page 6, please include the Cronbach's alpha coefficient information.

Reviewer 2 Report

Comments and Suggestions for Authors

The manuscript “The Influence of Mindful Parenting on Children's Creative Tendency: The Chain Mediating Effect of Parent-child Intimacy and Connectedness to Nature” presented an interesting study to explore the influence of parenting environment and natural environment on creative tendency. However, several significant issues remain to be considered:

1. I am not sure what this sentence (line 35-36) means in the first paragraph: “This creativity is mostly a prospective and personalized expression, nonetheless, as a result of children’s weak cognitive development.” This sentence mentions cognitive development, but the following sentences discuss environmental factors that affect children’s creative tendencies. What is the function and logic of this sentence in this paragraph?

2. Perhaps the most significant issue that needs to be solved lies in the Introduction. The evidence about the relationship mindful parenting and children’s creativity tendency is too scarce and unclear. Now that the benefits of mindfulness parenting have been enumerated, is there a more direct relationship or evidence to children’s creative tendency? A more detailed introduction to the relationship mindful parenting and creativity tendency is recommended. Besides, the introduction fails to place the work in relation with previous literature on the topic, some sort of transition would be beneficial. The authors can clarify whether the relationship between mindful parenting and creativity tendency has been demonstrated before.

3. Is there any other theoretical support for the variable (connectedness to nature) in the relationship between mindfulness parenting and creativity tendency? In addition, the inference of the mediating role of this variable is not convincing. It is recommended to revise this part of the writing, and add some more relevant research and sort out the writing logic to support the relationship between variables.

4. In general, the context in the Introduction is concise, but the content is not enough to clarify and support the research. The explanation of the relationship between variables, and the connection or difference with previous relevant studies are not specific enough and persuasive enough, and the highlights of this study are not reflected.

5. Please provide the age range of the children’s mothers who were recruited.

6. For the measure on every variable, I believe at least one example item is needed.

7. Line248: The 0.060 value of CI: [0.030, 0.060] in the paragraph is different from the value given in Figure4, which corresponds to 0.059.

8. The discussion of what was written in this section feels like a supplement to the introduction. However, the main purpose of the discussion part is to compare and analyze the similarities and differences, advantages and disadvantages between this paper and the past literature in terms of topic selection, methodology, results, etc., and based on these analyses to draw out new ideas, conclusions, and to explore new laws. For example, in the section 4.1 (Line260) mentions “However, there is still some debate regarding the connection between mindful parenting, a crucial aspect of healthy nurturing, and children's creative tendency.” So, what exactly is that debate? Therefore, it is recommended that the section be reorganized.

9. The highlights of this research is not reflected in the article. Mindfulness parenting may be a recent research hotspot, put together with creativity tendency to do research, has anyone done it before? The article does not explicitly mention that it is only mentioned as a supplement to previous studies and gives theoretical support in this regard. It's unclear how this study is novel or how it moves the field forward.

10. The citation of recent years’ research literature is relatively small. It is recommended to improve the reading of the latest relevant research and expand the content in the corresponding section.

11. The full text only used a theoretical model: ecosystem theory, will it not be enough or too little ? There are four variables in research.

Comments on the Quality of English Language

Extensive editing of English language required

Reviewer 3 Report

Comments and Suggestions for Authors

Overall, I thought this was an interesting paper that makes a meaningful contribution to the literature. The methodology, results and discussion were appropriate (and I commend the work involved in this). However, I do think a couple of areas of the manuscript could be improved before this is accepted for publication. Please see below for specific comments:

Introduction

1)    It would be of benefit to the paper to explain what a ‘chain mediation model’ is when it is first mentioned in lines 107-108. 

2)    In the first paragraph of the introduction, it would be good to explain why the study is of importance. Why should we know more about the development of childhood creativity?

Methods

3)    The phrasing on lines 153-155 could be clarifies, in particular, what does the phrase “Parents who have knowledge of their preschool children's situation” mean?

Discussion

4)    I would suggest swapping the order of 4.5 and 4.4 so that you discuss limitations before the overall contributions. 

Round 2

Reviewer 2 Report

Comments and Suggestions for Authors

The study is well improved. However, some clarifications are needed.

1. In the first edition of the paper, the content of the relationship between parent-child relationship and creativity tendency is not clear enough in the Introduction (Line70-81). Besides, this part of the reference is too old, the three are references of the 19th century.

2. In the Response to Comment 3: “Connectedness to nature refers to an intimate interaction between humans and nature in which people are emotionally attached to nature, cognitively blend nature with themselves, and physically experience nature 's allure and are willing to embody with it [31]. The expression of this sentence is not clear, and the following sentence (congnitively ...) may be Chinglish.

3. In the Response to Comment 2: “Based on the above re-search, this study proposes Hypothesis 1: Mindful parenting can positively predict children's creativetendencies.” The word “re-search” in this sentence may be misspelled.

4. “the chain mediation effect of parent-child intimacy and connectedness to nature: Mindful parenting Parent-child intimacy Connectedness to nature Children's creative tendency (a1*d*b2, mediating effect = 0.043, SE = 0.008, bootstrap 95% CI: [0.030, 0.059], accounting for 0.10 of the total effect).” Does the 0.01 satisfy the requirement?

Comments on the Quality of English Language

Extensive editing of English language required

Round 3

Reviewer 2 Report

Comments and Suggestions for Authors

For Comment 4, the authors respond that “We have adjusted the value from 0.10% to 9.91%”, but how to adjust? Why the other effects have not changed? Such an adjustment is too non-standard.
